# Chayote Fruit (*Sechium edule* var. *virens levis*) Development and the Effect of Growth Regulators on Seed Germination

**DOI:** 10.3390/plants12010108

**Published:** 2022-12-26

**Authors:** Yeimy C. Ramírez-Rodas, Ma. de Lourdes Arévalo-Galarza, Jorge Cadena-Iñiguez, Ramón M. Soto-Hernández, Cecilia B. Peña-Valdivia, José A. Guerrero-Analco

**Affiliations:** 1Colegio de Postgraduados, Campus Montecillo, Km. 36.5 Carretera México-Texcoco, Montecillo 56230, Mexico; 2Colegio de Postgraduados, Campus San Luis Potosí, San Iturbide No. 73, Salinas de Hidalgo, San Luis Potosí 78600, Mexico; 3Red de Estudios Moleculares Avanzados, Clúster Biomimic^®^, Instituto de Ecología, A. C. Carretera Antigua a Coatepec 351, Xalapa, Veracruz 91073, Mexico

**Keywords:** sigmoid growth, endosperm, endocarpic fruit, recalcitrant seed, transpiration, viviparism

## Abstract

The chayote fruit is a nontraditional vegetable belonging to the Cucurbitaceae family. The fruit has an endocarpic recalcitrant seed that emerges postharvest, drastically shortening its shelf life. In this study, the changes during fruit and seed development before and after harvest (ah) are reported. Additionally, in order to investigate how growth regulators (GRs) affect seed germination, 2-cloroethylphosphonic acid (CPA) (200 µL L^−1^), gibberellic acid (GA_3_) (100 and 200 mg L^−1^), auxin (2,4-D) (0.5 and 1.0 mM), and abscisic acid (ABA) (0.5 and 1.0 mM) were applied after harvest. The results showed that the chayote fruit reached horticultural maturity at 21 days after anthesis, with a sigmoid trend: phase I featured slow growth and high transpiration; in phase II, growth was accelerated and accumulation of endosperm was observed; and in phase III, both growth rate and transpiration were reduced, soluble sugars increased, and the seed showed 25% cotyledon development. At day 13 ah, CPA, GA_3_, and 2,4-D (0.5 mM) increased seed germination, with values between 10 and 15 mm of the embryonary axis, and the treatments with 2,4-D (1 mM) and ABA (0.5 and 1.0 mM) retarded their growth (2–6 mm). This research allowed us to reveal the phenological phases and the shelf life of the chayote fruit, as well as the results of possible postharvest treatment with GRs; our results suggest that strategies to delay viviparism and prolong the shelf life of the fruit should be applied before 10 days ah, when the embryonic axis of the seed has not developed.

## 1. Introduction

The chayote (*Sechium edule* Jacq. Sw.) belongs to the Cucurbitaceae family. The fruit has economic importance as a nontraditional vegetable. Currently, the main markets for chayote are Mexico, the USA, Canada, Japan, and some European countries [1], although Mexico and Costa Rica are the main exporters, with market shares of 53 and 47%, respectively. The *virens levis* variety is the most important [2,3].

The attractiveness of the chayote fruit is due to its low caloric content (13–19 kcal 100 g^−1^ fw) and high moisture content. Its highly varied nutritional composition includes fiber (0.4–7.53%), soluble sugars (1.66–3.30%), amino acids (12.31 mg g^−1^ of protein), and minerals such as potassium, calcium, phosphorus, magnesium, sodium, and zinc. The fruit is also rich in vitamins C (7.70–18.1 mg g^−1^) and E (0.12 mg g^−1^), and folic acid (93 µg g^−1^) [4]. The fruit has a short shelf life at room temperature (~10 d), mainly due to disease incidence [5] and premature germination of the seed inside the fruit (viviparism) [3]. In this regard, there are no studies that describe the changes that occur during the development of the fruit and in its seed until viviparism appears. The chayote, like other species of the Cucurbitaceae family, is harvested at the horticultural stage between 18 and 21 days after anthesis (daa) [6]; cucumber (*Cucumis sativus* L.), for example, is harvested from 12 to 20 daa [7,8].

The chayote seed is endocarpic and recalcitrant. These seeds are sensitive to dehydration and are vulnerable to loss of viability soon after harvest [9,10]. They have a very active metabolism and a low protein accumulation in the vacuoles, which causes their low resistance to desiccation [10,11]. When the chayote fruit is harvested at horticultural maturity, its seed development is not interrupted and the germination process can begin, particularly if the fruit is exposed to some preharvest or postharvest stress [12]. For example, if the shoots from which the fruits hang are damaged or withered, chayote germination is induced; moreover, if the fruits are stored at low temperatures (7 °C), viviparity occurs earlier and in a higher percentage of fruits than those kept at room temperature [5].

Growth regulators are very important in seed development, and abscisic acid (ABA), gibberellins (GAs), auxins, and ethylene, among others, play important roles in the germination process [13,14]. ABA regulates the seed’s reserve accumulation, latency, and germination inhibition [15]; auxins play an important role in latency. When ABA levels are low, the GA levels increase, allowing germination to begin via effects on the tissues surrounding the radicle (endosperm and testa) and stimulation of the expansion of the embryonary cells [16,17]. In addition to GA, ethylene acts as an antagonist for ABA and promotes germination, breaking latency. Under stress, the tissues synthesize higher amounts of ethylene and induce radicle growth [14,17].

The germination of viviparous seeds inside the fruit also occurs in papaya (*Carica papaya* L.) [18], sapote (*Pouteria sapota*) [19], and tomato (*Solanum lycopersicum*) [20]; it does not affect the appearance of these fruits, since the seeds remain inside. In contrast, when the seed germinates in chayote, it is exposed in the distal part of the fruit and causes the fruit’s deterioration by allowing the entry of pathogens, affecting the fruit’s appearance and reducing its shelf life. Therefore, the aim of this study was to describe the stages of development of the chayote fruit and seed in var. *virens levis* during pre- and postharvest periods, and the effects of four growth regulators applied to chayote fruits soon after harvest and evaluated 13 days after, in order to contribute to the implementation of strategies to delay viviparism and increase the shelf life of these fruits.

## 2. Results and Discussion

### 2.1. Fruit and Seed Description

The *virens levis* fruits are pear-shaped at harvest. The epicarp of this variety is 0.46 mm thick, light green, and has a color value of L = 84.60, C = 52.37, °H = 112.43; the mesocarp has a light green color, L = 95.7, C = 37.15, °H = 113.51, and a white endocarp; the fruit is spineless and contains only one seed (Figure 1A). The fruit size ranges between 9.8 and 11.53 cm in length and 6.79 and 8.33 cm in width, with an average weight of 379 g per fruit. The chayote seed is located near the distal part of the fruit, and when it is fully developed, its size is 3.4 cm in length. The seed is composed of two cotyledons and a 1.50-mm-thick soft testa that surrounds the embryo (Figure 1).

### 2.2. Fruit Development

The most commonly used variables to evaluate the growth of fruit are the length, width, and dry matter accumulation [21,22]. After anthesis, the chayote fruit begins its development from an inferior ovary and growth follows a simple sigmoid-type curve, comprised of three phases, like that of other cucurbit fruits such as pumpkin, cucumber, and chilacayote [8,23,24]. In phase I, the fruit and the seed showed slow growth (cell division). This occurred up to 9 daa, at which point the fruit had reached about 41% of its final size (final size = 21 daa), while the size of the seed was around 33% (Figure 2 and Figure 3A,B). In cucurbits such as cucumbers, during the cell division phase, the number of cells in the fruit increases ~200% during the first 3 daa and the fruit develops spines at 4 daa [25]. In pumpkins, the fruit completes cell division and initiates the differentiation of the embryo into the seeds, as well as the accumulation of endosperm and expansion of the covers [26].

During phase II, the growth of the fruit and seed of the chayote was exponential up to 18 daa, particularly between days 12 and 15, when the fruit showed an increase of 24% and the seed of 60% (Figure 2 and Figure 3A,B). The endosperm in this phase was abundant, with a transparent gelatinous layer that expanded as the seed developed and reduced as the cotyledons reached full development. Between 15 and 18 daa, the development of cotyledons was observed to pass through the heart and torpedo stages characteristic of dicotyledonous seeds [27]. 

The accumulation of the gelatinous endosperm in the seed (Figure 2) occurs after the degeneration of the nucellus, leaving a cavity inside the ovule that is subsequently filled by the endosperm and finally by the embryo [28]. It has been reported that the endosperm is capable of detecting environmental changes and can produce and secrete signals that allow it to regulate the growth of the embryo [29]. In cucumber, an exponential increase in the cell size (~30 times) occurs between 5 and 16 daa; there is also an increase in the speed of expansion of the placenta (gelatinous) and the pericarp, and the aging of the spines [25]. In pumpkin, the fruits and seeds reach their maximum size near the end of this phase, registering a greater accumulation of starch in the mesocarp [26]. 

In both crops (cucumber and pumpkin) there is a thickening of the cuticle with greater epicuticular wax deposition, which allows the presence of a hydrophobic layer and initiates seed development. This is observable via fully expanded covers, hardening of the testa and development of cotyledons, and depletion of reserves such as starch and pectin [25,30]. In this sense, in cucumber fruits, the participation of five genes in rapid cell division has been mentioned (Csa3M062600, Csa4M002000, Csa6M499030, Csa2M250930, and Csa5M157410), and two have been reported to be related to exp onential cell expansion (Csa7M446860 and Csa1M495290) [31].

Phase III (18–21 daa) showed a reduction in the growth rate of the chayote fruit and seed, reaching lengths of 11.53 and 3.47 cm, respectively. The seed was not morpho-physiologically mature at this stage, since the cotyledons showed 25% development and other components of the embryo were not yet evident (Figure 2 and Figure 3A,B). 

In phase III of development in some varieties of pumpkin, the carotenoids responsible for the yellow or orange color of the fruits accumulate and the firmness of the epicarp and mesocarp decreases via the action of enzymes that modify the structure of the cell wall, which is related to the decrease in starch concentration (18.88%) and the increases in dry matter (16%), crude fiber, sugars, and soluble solids. Finally, the seeds reach their maximum size and are filled with fully developed cotyledons [25,32].

During chayote development, the initial fruit fresh weight (FW) in phase I ranged from 0.263 to 5.60 g, and the dry weight (DW) from 0.018 to 0.30 g. Therefore, between 15 and 18 daa (phase II) the average fresh weight increased by 73% and the DW increased by 77% compared to the previous phase. In phase III, the average weight was 379 g per fruit, and DW varied from 4% to 7%, with an average moisture content of 94.5% (Figure 3C,D).

The accumulation of fructose was similar to that of glucose and no sucrose was detected. In phase II, the concentration of fructose increased by 28% between 9 and 15 days; glucose increased by 21% in the same period. No significant differences in the hexoses were observed from day 15; the average values recorded were 1.04% for fructose and 0.76% for glucose (Figure 4), similar to the results reported for different chayote varietal groups such as *nigrum spinosum, virens levis*, and *nigrum xalapensis* with values of 2.03 and 2.42%, respectively [5,33].

Transpiration depends on factors such as the permeability of the epidermis of the fruit, temperature, radiation, and relative humidity, among other factors. The maximum rate of transpiration can be recorded just after the setting of the fruits and declines rapidly, reaching its minimum in the middle of the growth period; the minimum transpiration rate reached by fruits depends on the species [34]. In chayote, the tiny young fruits had high transpiration values (22–40 g kg^−1^ h^−1^) that reduced as the fruit grew, reaching values between 1.1 and 2.7 g kg^−1^ h^−1^ (Figure 5). 

Even if fruits were harvested before 15 daa, their shelf lives would be shorter than those of larger fruits due to their higher metabolic activity and the greater dehydration of the fruit. In apricot fruits (*Prunus armeniaca* L. cv Tyrithos), the maximum transpiration rate of ~0.55 mmol m^−2^ s^−1^ was recorded in young fruits of 6 daa and this rapidly decreased to ~0.35 mmol m^−2^ s^−1^; in kiwifruit (*Actinidia deliciosa* var. Deliciosa), transpiration had a maximum value of 2.3 mmol m^−2^ s^−1^ in the first days of fruit setting, after which it decreased rapidly and in the final phase of development it reached values close to 10% of the initial value. The decrease in transpiration during fruit development has been associated with the accumulation of epicuticular waxes in the fruit that act as a barrier to reduce moisture loss [34,35].

In orange (*Citrus sinensis*) fruits, weight loss decreases as the epicuticular waxes increase, principally during the first phase of development (60 days after flowering, daf) and which are distributed in the form of flattened plates that became larger and dense at 180 daf [36]. During the development of the chayote fruit, it was observed that from 12 daa, the fruits had a smoother and brighter epidermis that denoted the presence of epicuticular waxes (Figure 2 and Figure 5).

### 2.3. Seed Development during Postharvest Period

Once the fruit was harvested (horticultural maturity), the seed registered about 25% of the development of the embryo (cotyledons) (Figure 6) and was in the initial phase of maturity of dicotyledonous seeds, denoted by the heart and torpedo stages [37]. The seed continued to grow for a further 7 d and reached 3.4 cm in length, while the cotyledons reached morphophysiological maturity (cotyledons fully developed) at 10 dah. The cotyledons covered the seed cavity and there was an absence of endosperm. The endosperm and cotyledons contained the highest proportions of reserve substances, the mobilization of which supports the growth of the new seedling until it is photosynthetically active [28]; therefore, the endosperm was probably reabsorbed by the cotyledons to function as a reserve organ. The cotyledons followed a similar pattern; at the time of harvest, 18–25% of their development was recorded and 7 days after that they reached 85% of their final size, covered the cavity of the seed completely, and an absence of endosperm was observed. At approximately day 10 after harvest, the seed reached its optimum morphophysiological stage and was able to germinate.

This indicated that before 13 dah, the growth of the embryonic axis (GEA) began from the cotyledons towards the distal part of the fruit, causing the opening of the testa and consequently the fruit. Between days 13 and 15, the GEA ranged between 8 and 12 mm, while on day 18 it reached 16 mm. Additionally, the plumule and in some cases the radicle were clearly observed. From 13 dah, chayote fruits lost commercial quality, as the embryonic axis was exposed in the distal part of the fruit by about 8 mm as a product of germination. The plumule was observed at 15 dah, as was the radicle of the new plant in some cases (Figure 6). 

The chayote seed is recalcitrant; this type of seed does not tolerate dehydration, so a loss of 50% of its moisture affects its viability and the absence of dessication resistance in recalcitrant seeds is attributed to the lack of late-embryogenesis proteins (LEA) [9]. On the other hand, in orthodox seeds the presence of LEA proteins favor their tolerance to moisture loss [10]; for example, Delahaie et al. [11], comparing the proteome of two legume species (Fabaceae), reported the absence of LEA proteins in the recalcitrant seeds of Australian chestnut (*Castanospermum australe*) compared to the orthodox seeds of barrel clover (*Medicago truncatula*). These proteins are hydrophilic and function by replacing water molecules by forming hydrogen bonds with other proteins, acting as protectors to prevent cell death in the period of dehydration [38].

A mechanism of seed protection against dehydration and loss of viability is the presence of covers such as the epicarp, in addition to phenolic compounds [37]. It is also possible that there are many other molecular protective mechanisms that could be missing in recalcitrant species such as antioxidant defenses, nonreducing sugars, heat shock proteins, and/or induction of cell wall modifications [11]. The chayote seed possesses a thin testa of 15 mm, which the embryo can break easily. Another cover that surrounds the seed is the pericarp, which provides the required humidity for a while.

### 2.4. Postharvest Growth Regulators and Their Influence on Germination

The release of ethylene by 2-cloroethylphosponic acid (CPA) in chayote fruits significantly increased the size of the embryonary axis, up to 50% of the basal opening of the fruits; following treatments with 2,4-D and ABA (1mM), viviparity was observed in 33% and 16% of the fruits, respectively, and in the ABA (0.5 mM) treatment, the basal opening was not perceptible after 13 days of storage (Figure 7). The complex interactions between growth regulators during germination depend on the synergy or antagonism between them. Even small dose changes provoked different results, such as the auxin (2,4-D) treatment at 0.5 mM showing higher rates of viviparism than the 1mM treatment [39].

In wild chayote fruits, it has been observed that the rate of germination and emergence of the seed outside the fruit is almost null, a phenomenon that can be attributed to the high content of phenols and triterpenes such as cucurbitacins [2]. Additionally, some growth hormones dependent on the terpene pathway are involved in seed germination, such as ABA, which regulates different aspects of development and the beginning of embryogenesis, as well as the maintenance of dormancy and germination [13]. Indole-3-indoleacetic acid, the most abundant auxin in plants, interacts synergistically with ABA, inducing genes that are necessary for ABA synthesis [40]. Gibberellins, on the other hand, promote germination by competing with the levels of abscisic acid; in addition, they are necessary for the weakening of the testa that surrounds the radicle [16]. Finally, ethylene is a promoter of germination. For this reason, in recalcitrant seeds, such as chayote, the increase or decrease in the levels of antagonistic hormones is what allows the seed to germinate [5,41]. 

Figure 8 summarizes the results of this study. The phenological phases of the chayote fruit begin with anthesis, followed by growth, which includes the period between 18 and 21 daa; the fruit is harvested at horticultural maturity and the seed continues to develop, reaching morphophysiological maturity at 10 dah. Being a nonclimacteric fruit, senescence begins after harvest, and because of the early seed germination, fruit dehydration and disease problems shorten the shelf life of the fruit. Knowledge about the changes that occur during the development of the fruit and seed in chayote will allow the implementation of technologies to increase the shelf life of this fruit.

## 3. Materials and Methods

### 3.1. Fruit Development

The chayote orchard in the present study is located in the National Germplasm Bank of *S. edule* in Mexico (BANGESe), with vegetation of the mesophyllous mountainous forest (cloud forest) type at 1340 m altitude with an average annual temperature of 19 °C, RH of 85%, and annual rainfall of 2250 mm. Female flowers *of Sechium edule* var. *virens levis* were selected and marked, then the fruits were harvested at 3, 6, 9, 12, 15, 18, and 21 days after anthesis (daa) and those free of mechanical damage and diseases were selected. 

The fruit growth was recorded as the length of fruit and seed, using a digital vernier caliper (Steren^®^, Model Her-411, CDMX, Mexico) with a precision of 0.1 mm. The weight was recorded on an analytical balance with a precision of ±0.001 g (Scientech, SA410IW, Boulder, CO, USA). The development of the cotyledons was reported in percentages, for which a scale was established (Figure 1D). Transpiration was estimated via the changes in the weight of the fruit through time, and the measurements were recorded for consecutive hours on an analytical scale with a precision of ± 0.001g (Scientech, SA410IW, Boulder, CO, USA). 

The color of the epicarp and mesocarp was measured with a colorimeter (3NH Technology Co., Ltd., nr20xe, Shenzhen, P.R., China) on the equatorial region of the fruit; the results were expressed in values of luminosity (L), saturation or chroma (C), and tone or hue (°H).

Soluble sugars were quantified using high-performance liquid chromatography (HPLC) coupled to a refractive index detector (Perkin Elmer™, Series 200, Shelton, CT,USA). The evaluation was made using samples of 3 g of pericarp, finely chopped, from the equatorial region of the fruit. The alcoholic extraction and solid-phase extraction (SPE) were based on the method of Macherey-Nagel [42]. A Pinnacle II Amino 5 µm, 150 × 4.6 mm column (Restek™, Bellefonte, PA, USA) was used. The mobile phase under isocratic conditions was acetonitrile/water (80:20, *v*:*v*) with a run time of 14 min. Calibration curves were prepared for fructose, glucose, and sucrose (99.5%, Sigma-Aldrich, (St. Louis, MO, USA); standard solution mixtures of 4.87 mg mL^−1^, 5.92 mg mL^−1^, and 4.69 mg mL^−1^, respectively, were prepared in methanol:water (1:9, *v*:*v*) and dilutions of fructose from 0.0381 to 2.435 mg mL^−1^, glucose from 0.0421 to 2.6488 mg mL^−1^, and sucrose from 0.0333 to 2.0985 mg mL^−1^ were made for the calibration curves. Chromatograph conditions were as follows: column temperature of 35 °C, flow rate of 1 mL min^−1^, and injection volume of 10 µL. The results were expressed as g 100 g^−1^ fw.

### 3.2. Seed Development 

Fruits were harvested at horticultural maturity (18 ± 2 daa), with an average size of 9.8–11.2 cm in length and 6.9–7.9 cm in width, and kept at room temperature (24 ± 1 °C; 65 ± 1% RH) for 0, 7, 10, 13, 15, and 18 days. On each test day, three fruits were cut in half to measure the length and width (cm) of the seed and cotyledons using a digital vernier caliper (Steren^®^, Model Her-411, Mexico City, Mexico) with a precision of ± 0.1 mm. The development of the cotyledons (%) (DC) was recorded according to the scale described in Figure 1D. The growth of the embryonic axis (mm) (GEA) was quantified from the breaking of the testa at the distal part of the fruit. 

### 3.3. Application of Growth Regulators in Postharvest Period

Fruits at horticultural maturity (18 ± 2 daa) were kept at room temperature (24 ± 1 °C; 65 ± 1% RH) and evaluated at 13 days after harvest. Four growth regulators were applied to chayote fruits: 2-chloroethyl phosphonic acid (200 µL L^−1^) (CPA, ethylene releaser), gibberellic acid (GA_3_, 100 and 200 mg L^−1^), 2,4-dichlorophenoxyacetic (2,4-D; 0.5 and 1.0 mM), abscisic acid (ABA, 0.5 and 1 mM), and a control (water). The fruits were submerged in each aqueous solution at room temperature (24 °C) for 5 h, dried, and placed in plastic boxes. The variables evaluated were as follows: exposed seed in the distal part of the fruit (viviparism) (%) and growth embryonic axis (GEA, mm) at 13 dah. Each fruit was a replicate, and each treatment had six fruits.

### 3.4. Experimental Design and Statistical Analysis

The experimental design was completely randomized, with the fruit as the experimental unit. The growth curves were fitted to a logistic model using 10 replicates at each evaluation stage of fruit and seed growth, as well as for accumulation of fresh and dry weight where Y = a/(1 + be^−cx^), where Y = response variable, a = maximum value of Y, b = (Ymax − Ymin)/Ymin, c = percentage growth increase of the model, x = time (independent variable), and e = Euler value. The fitted curves were obtained using the CurveExpert Basic 2.2.3 program [43] with a significant R^2^ correlation coefficient. Since the values of transpiration did not present normality, the transformation of their data was carried out using Box-Cox (λ = −0.3434). For each evaluation of sugars and transpiration variables, three replicates were performed, while for the growth of the embryonic axis, six replicates were used per treatment. Data were analyzed via a one-way ANOVA, and Tukey’s post hoc test (α = 0.05) was also performed. R-Studio version 4.1.0 program [44] was used.

## 4. Conclusions

This study described the behavior and changes of chayote fruit from anthesis to postharvest and the seed viviparity. The growth curves of characteristics such as the length and width of the fruit and the seed, and the fresh and dry weight of the fruit, were similar to those of other cucurbits of agronomic interest (cucumber, pumpkin, and chilacayote), and indicated that the development of the fruit and the seed of the chayote var. *virens levis* followed a sigmoid trend, with slow growth in the first 9 daa, exponential growth between 9 and 18 daa, and slow growth between 18 and 21 daa. Once the development of the cotyledons of the seed had begun (15 daa), the seed reached germination (viviparism) in the following 16 d. The application of growth regulators such as CPA, GA_3_, and 2,4-D (0.5 mM) accelerated seed germination, but the application of auxin 2,4-D (1 mM) and ABA delayed it. In order to prolong the shelf life of these fruits, it is suggested that strategies that contribute to delaying viviparism in chayote, such as low temperatures or growth retardants, should be applied before 10 dah, when the embryonic axis of the seed has not yet developed.

## Figures and Tables

**Figure 1 plants-12-00108-f001:**
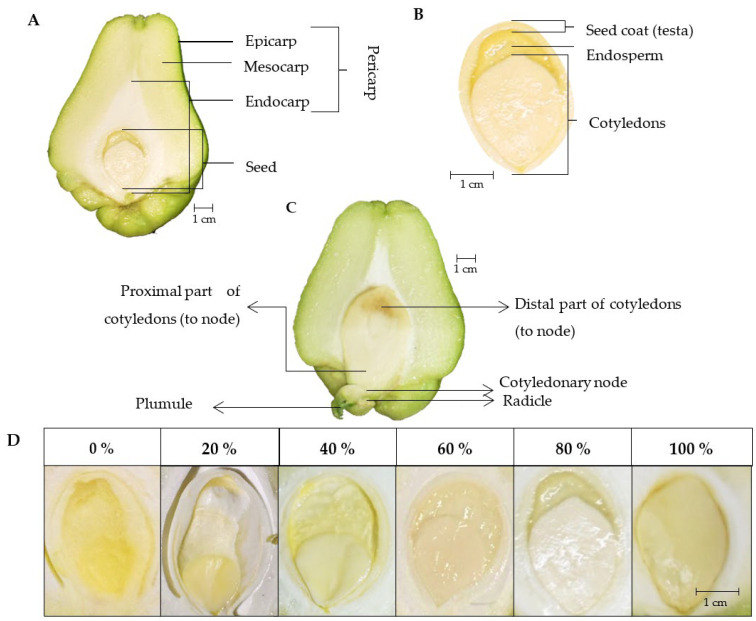
Morphology of the fruit and seed of chayote *Sechium edule* var. *virens levis* at horticultural maturity (18 ± 2 days after anthesis). (**A**) Fruit description; (**B**) seed components; (**C**) fruit with germinated seed and embryo; (**D**) scale of cotyledon development.

**Figure 2 plants-12-00108-f002:**
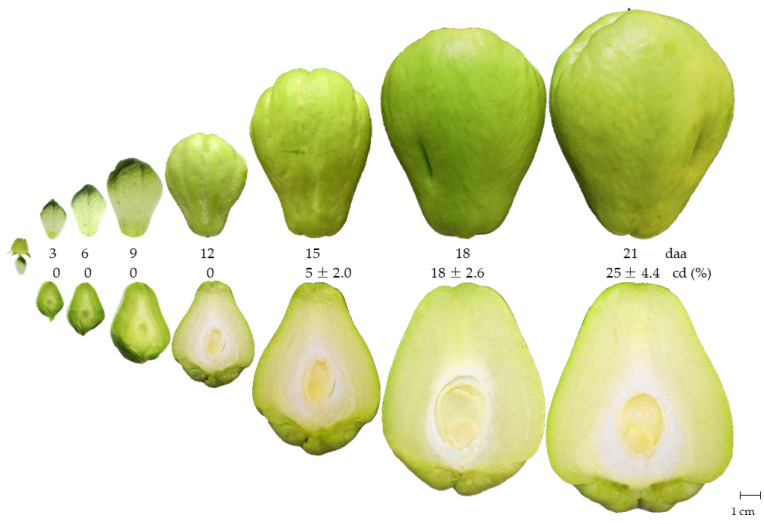
Fruit and seed development of chayote *Sechium edule* var. *virens levis* at 3, 6, 9, 12, 15, 18, and 21 daa = days after anthesis; cd = cotyledon development (*n* = 10 ± standard deviation).

**Figure 3 plants-12-00108-f003:**
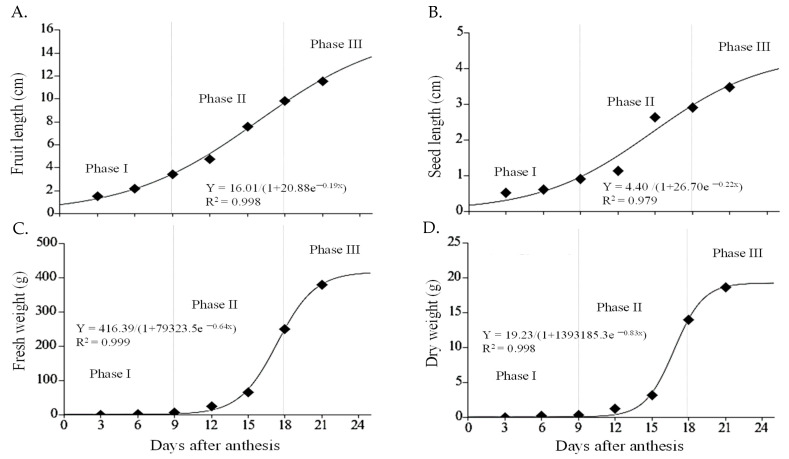
Fruit and seed development of chayote *Sechium edule* var. *virens levis*. Fruit (**A**) and seed (**B**) growth curves (*n* = 10); accumulation of fresh (**C**) and dry weight (**D**) (*n* = 3) during the development of chayote fruit *virens levis* 3, 6, 9, 12, 15, 18, and 21 days after anthesis.

**Figure 4 plants-12-00108-f004:**
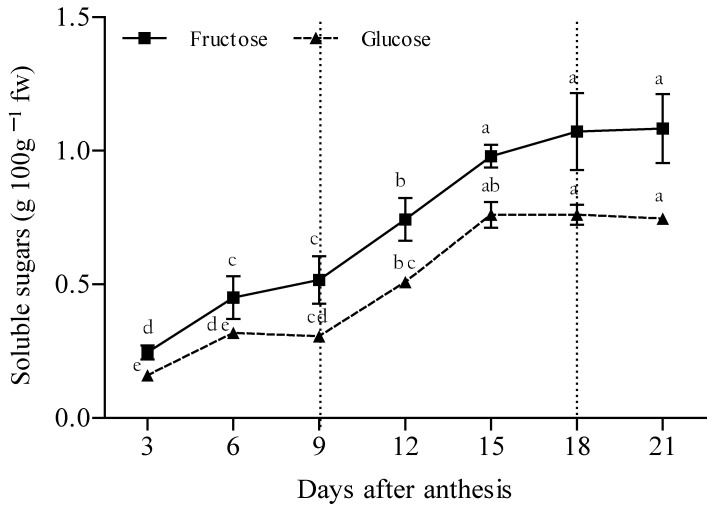
Accumulation of soluble sugars in fruits of chayote *Sechium edule* var. *virens levis* (*n* = 3 ± standard deviation). Different letters show significant differences in the same sugar (Tukey test, α = 0.05).

**Figure 5 plants-12-00108-f005:**
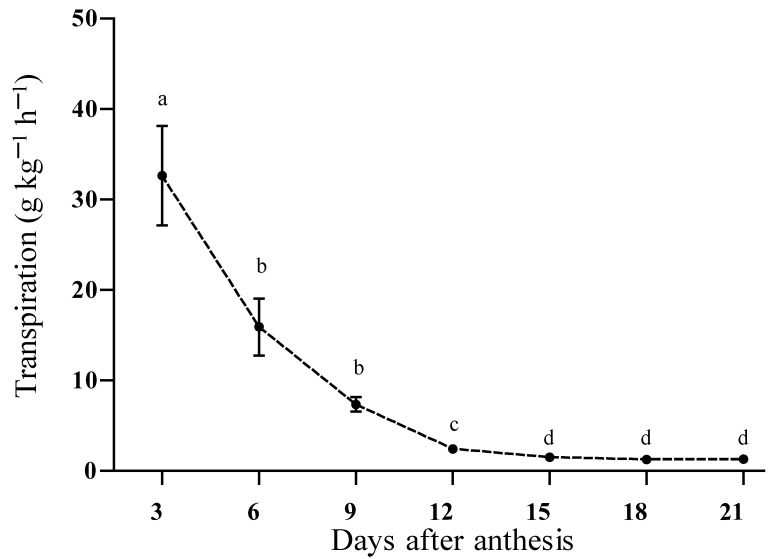
Transpiration in chayote fruits *Sechium edule* var. *virens levis* during development (*n* = 3 ± standard deviation). Different letters show significant differences (Tukey test, α = 0.05).

**Figure 6 plants-12-00108-f006:**
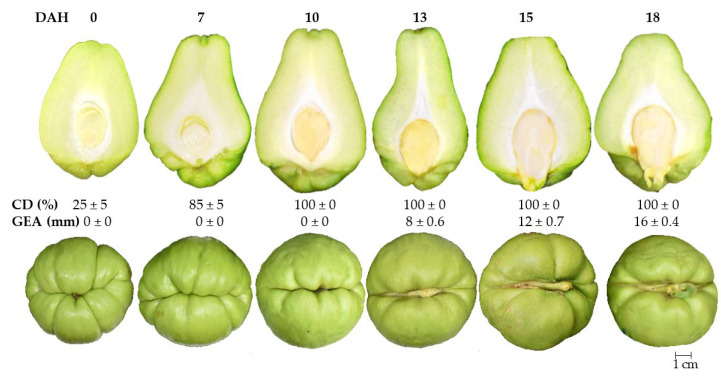
Postharvest characteristics of seed development, cotyledons, and embryonic axis of chayote *Sechium edule* var. *virens levis*. DAH = days after harvest; CD = cotyledon development; GEA = growth embryonic axis (*n* = 3 ± standard deviation).

**Figure 7 plants-12-00108-f007:**
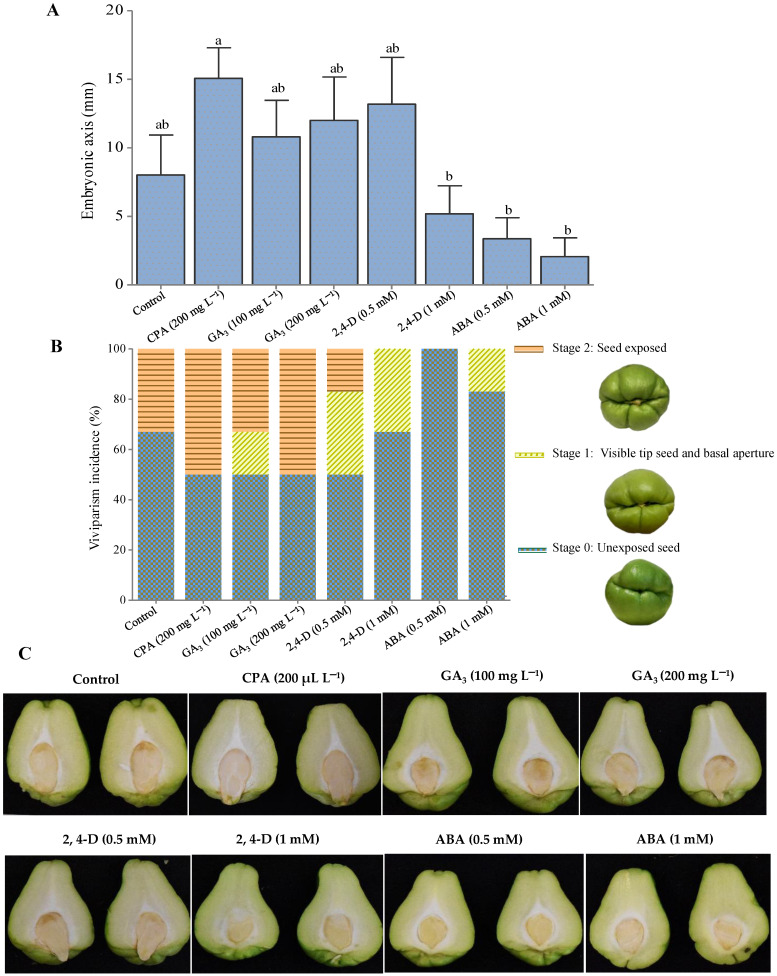
Embryonic axis growth (**A**) and viviparism incidence (**B**) in *Sechium edule* var. *virens levis* fruits after 13 days of storage at 24 °C and 65 ± 1% RH, following treatment with growth regulators (**C**): Control (distilled water); 2-chloroethyl phosphonic acid (CPA, 200 µL L^−1^); gibberellic acid 3 (GA_3_, 100 mg L^−1^); GA_3_ (200 mg L^−1^); 2,4–dichlorophenoxyacetic (2,4-D, 0.5 and 1 mM); abscisic acid (ABA, 0.5 and 1 mM). *n* = 6 ± standard deviation. Different letters show significant differences (Tukey test, α = 0.05).

**Figure 8 plants-12-00108-f008:**
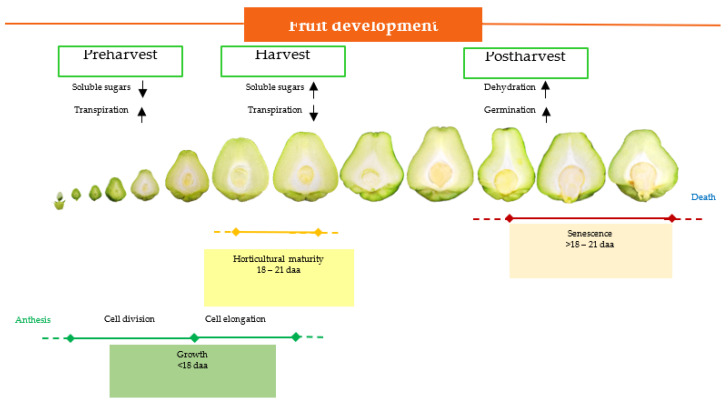
Physiological phases of fruit development of chayote *Sechium edule* var. *virens levis*. Solid lines indicate intensity (
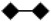
) and dashed lines indicate weakness (---); the 
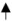
 arrow indicates high values and the 

 arrow indicates low values.

## Data Availability

Not applicable.

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
