# Peer review of "Chayote Fruit (Sechium edule var. virens levis) Development and the Effect of Growth Regulators on Seed Germination"

_plants, 2022, doi:10.3390/plants12010108_

Round 1

Reviewer 1 Report

Overall, there is little scientific information on chayote squash. The authors present a manuscript that seems to be a review then reverts into presentation of data on growth curves, vivipary, and some growth regulator work. This is where some scientific problems come up. For instance,  the number of fruit sampled per stage, or for gr work, is never presented, either in text or in figures or tables. This led me to the next question, which is the percent data in Figure 7. No standard deviations are given for % vivipary although they are for the embryonic axis. No mention is made of the starch content in chayote, which most likely is present in immature stages. The soluble sugar method described does not address starch. 

The English needs some strong improvement. Sentences need appropriate punctuation and altered so that the key information is clear. 

Application of growth regulators to an edible crop, even in the early stages, may pose some regulatory questions for producers and consumers and should be considered in later research approaches.

Author Response

Reviewer 1:  Thank you for your suggestions to improve the manuscript, that the main objetive is to show the fruit development and critical time to take action to delay viviparism. So in order to improved the manuscrip we:

  1. Included the number of replicates used in each section was added in the Materials and Methods section (section 3.4) and in each image caption where was required.
  2. No statistical analysis is shown in the variable "Viviparity" because each replicate was a fruit, so we modified the graph for better understanding

3) The objetive of the growth regulators application was to know their relationship with viviparism and take further actions on that sense.

Reviewer 2 Report

This paper seems interesting, the chayote fruit is not traditional, but important for some countries. The design of this paper is not theoretically deep, but expressed important essential data and suggestions about this fruit. So I believe that this paper have great practical meaning, and can give guidance for the operation of this fruit. some place need more specific, like:

1. line 151, 'in this phase' means what phase, please specific in the new paragraph.

2. Figure 4, usually we start the biggest  date as 'a' in the difference significance analysis, but in figure 4, the author started from the smallest data, please change it.

3. Figure 7, only one set of data have difference significance analysis, but the other set is missing, please add it.

4. 3.3, line 343, the concentration of plant growth regulator, and the treatment time, how do author choice the certain parameter in paper?

5. the postharvest storage condition, please account in detail, like temperature, time.

Author Response

Thank you for your valuable comments, so according to the suggestions the descriptions of each phase are indicated in the correspondent paragraph.

  1. The significance of the data in Figure 4 was reordered, starting “a” with the highest data and we added the missing ones.
  2. No statistical analysis is shown in the variable "Viviparity" because each replicate was a fruit, so a modified graph is presented in order to make this variable more understandable.
  3. In section 3.3 the concentration of the regulator and the exposure time to the treatment were chosen according to previous unpublished trials.
  4. In Figure 7, the relative humidity data was added to complete the data for storage information.

Reviewer 3 Report

I am honored with privilege to review manuscript “Chayote Fruit (Sechium edule var. virens levis) Development  and the Effect of Growth Regulators on Postharvest”.

The merit of manuscript is reducing viviparisam and appearance embryonic axis in fruit by application of growth regulators. Although excellent results are achieved, those results could be result of biased fruit selection. As seen in picture 8, seed in fruits in horticulture maturity could be in various stages of development, and it lasts only 3 days. In control fruits, after 13 days (not 3 days) there is embrion appereance. To conclude – it is very hard to choose fruits at same stage of development, but solution is to have large number of fruits. Unfortunately, in section 3.3 there is no mention of fruit number per treatment.

Many experiments on fruits have questionable results because of inappropriate fruit selection. Please describe precisely how fruits were selected (having in mind equal maturity). Sometimes it is impossible to select fruit according to maturity (due to their small size e.g. berries), but since chayote is relatively large fruit, description of fruit selection must be included in material and method section. Just days after anthesis is not enough, since individual fruit development is depended on many factors which can cause unequal fruit maturity despite same development number of days.

Although statistics is present (figure 7) there is no chapter about statistical analysis. Also, why no analysis of viviparism is presesented?

Last part of title “and the Effect of Growth Regulators on Postharvest” is not appropriate. Usually postharvest relates on multiple analysis which describes overall difference in fruit storage. In case of this study, application of growth regulators is related only to viviparism and embryonic axis, so I suggest changing of title accordingly.

Considering fruit development, it is already well described in paper Fu, A., Wang, Q., Mu, J., Ma, L., Wen, C., Zhao, X., ... & Zuo, J. (2021). Combined genomic, transcriptomic, and metabolomic analyses provide insights into chayote (Sechium edule) evolution and fruit development. Horticulture research, 8.

Author Response

Thank you for the valuable comments. So according to them the horticultural maturity data was adjusted to the corresponding image in Figure 8. We have been working on chayote and the horticultural maturity is reached at 18 ± 3 days, that data is used by growers.

  1. In Materials and methods section 3.4, the information of the number of replicates per treatment that was used in section 2.4 was added.
  2. Additional information for the fruit selection in each experiment was described in the Materials and methods, considering the number of days after anthesis and the size of the fruit, to be accurate about the horticultural maturity.
  3. The statistical analysis chapter is found in section 3.4 together with the experimental design section.
  4. No statistical analysis is shown in the variable "Viviparity" because each replicate was a fruit, so we modified the graph in order to clarify.
  5. The title was modified from “….the effect of postharvest growth regulators” to “ the effect of growth regulators on seed germination.
  6. According to the suggested paper: Anzhen Fu, et al.,2021. Combined genomics, transcriptomic, and metabolomic analyzes provide insights into chayote (Sechium edule) evolution and fruit development, Horticulture Research, Volume 8, 35, https://doi.org/10.1038/s41438-021-00487-1.

Although the described the analysis of the metabolome and transcriptional occurring in the growth stages in chayote fruits, the authors don´t  discuss the issue of seed germination in chayote, that is the main problem in postharvest. Our work is focused in the morphological description on each  stage of fruit and seed, that can contribute to detecting the optimal time in which treatments can be applied to extend the shelf life of the fruits.

Reviewer 4 Report

This paper deals with the investigation of the fruit development and the effects of phytohormones on the fruit storage in chayote. The results presented here are solid.

I have only one point. In Figure 1, the authors show the appearance of the fruit. Could you describe the developmental stages when the authors photographed (days after flowering, immature, or mature?) in 1A to 1C?

Author Response

Thank you for your comments, 

  1. The request information for fruit maturity in Figures 1A-1C was added.